# ThriftyDAgger: Budget-Aware Novelty and Risk Gating for Interactive Imitation Learning

**Ryan Hoque**[1]**, Ashwin Balakrishna**[1]**, Ellen Novoseller**[1]**,**
**Albert Wilcox**[1]**, Daniel S. Brown**[1]**, Ken Goldberg**[1]

**Abstract:** Effective robot learning often requires online human feedback and interventions that can cost significant human time, giving rise to the central challenge in interactive imitation learning: *is it possible to control the timing and length of interventions to both facilitate learning and limit burden on the human supervisor?* This paper presents ThriftyDAgger, an algorithm for actively querying a human supervisor given a desired budget of human interventions. ThriftyDAgger uses a learned switching policy to solicit interventions only at states that are sufficiently (1) *novel*, where the robot policy has no reference behavior to imitate, or (2) *risky*, where the robot has low confidence in task completion. To detect the latter, we introduce a novel metric for estimating risk under the current robot policy. Experiments in simulation and on a physical cable routing experiment suggest that ThriftyDAgger's intervention criteria balances task performance and supervisor burden more effectively than prior algorithms. ThriftyDAgger can also be applied at execution time, where it achieves a $100\%$ success rate on both the simulation and physical tasks. A user study ($N = 10$) in which users control a three-robot fleet while also performing a concentration task suggests that ThriftyDAgger increases human and robot performance by $58\%$ and $80\%$ respectively compared to the next best algorithm while reducing supervisor burden. See https://tinyurl.com/thrifty-dagger for supplementary material.

**Keywords:** Imitation Learning, Fleet Learning, Human Robot Interaction

## 1 Introduction

Imitation learning (IL) [1, 2, 3] has seen success in a variety of robotic tasks ranging from autonomous driving [4, 5, 6] to robotic manipulation [7, 8, 9, 10, 11]. In its simplest form, the human provides an offline set of task demonstrations to the robot, which the robot uses to match human behavior. However, this offline approach can lead to low task performance due to a mismatch between the state distribution encountered in the demonstrations and that visited by the robot [12, 13], resulting in brittle policies that cannot be effectively deployed in real-world applications [14]. *Interactive imitation learning*, in which the robot periodically cedes control to a human supervisor for corrective interventions, has emerged as a promising technique to address these challenges [15, 16, 17, 18]. However, while interventions make it possible to learn robust policies, these interventions require significant human time. Thus, the central challenge in interactive IL algorithms is to control the timing and length of interventions to balance task performance with the burden imposed on the human supervisor [19, 18]. Achieving this balance is even more critical if the human supervisor must oversee multiple robots at once [20, 21, 22], for instance supervising a fleet of self-driving taxis [6] or robots in a warehouse [23]. Since even relatively reliable robot policies inevitably encounter new situations that must fall back on human expertise, this problem is immediately relevant to contemporary companies such as Waymo and Plus One Robotics.

One way to determine when to solicit interventions is to allow the human supervisor to decide when to provide the corrective interventions. However, these approaches—termed "human-gated" interactive IL algorithms [15, 16, 24]—require the human supervisor to continuously monitor the robot to determine when to intervene. This imposes significant burden on the supervisor and cannot

---

[1]AUTOLAB at the University of California, Berkeley
Correspondence to ryanhoque@berkeley.edu

5th Conference on Robot Learning (CoRL 2021), London, UK.

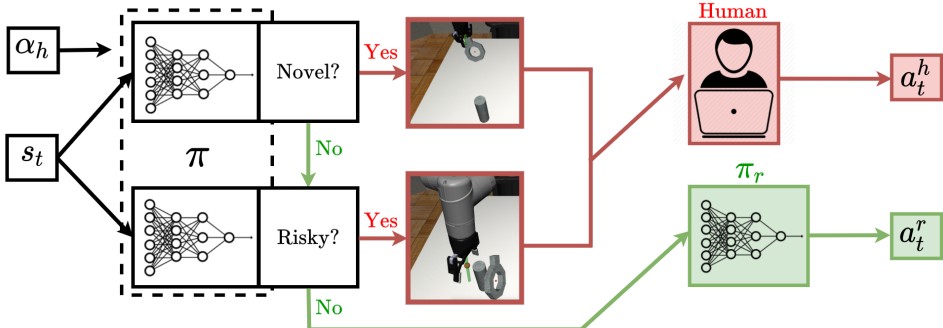

Figure 1: **ThriftyDAgger:** Given a desired context switching rate $\alpha_h$, ThriftyDAgger transfers control to a human supervisor if the current state $s_t$ is (1) sufficiently novel or (2) sufficiently risky, indicating that the probability of task success is low under robot policy $\pi_r$. Intuitively, one should not only distrust $\pi_r$ in states significantly out of the distribution of previously encountered states, but should also cede control to a human supervisor in more familiar states where the robot predicts that it is unlikely to successfully complete the task.

effectively scale to settings in which a small number of humans supervise a large number of robots. To address this challenge, there has been recent interest in approaches that enable the robot to actively query humans for interventions, called "robot-gated" algorithms [19, 25, 26, 18]. Robot-gated methods allow the robot to reduce burden on the human supervisor by only requesting interventions when necessary, switching between robot control and human control based on some intervention criteria. Hoque et al. [18] formalize the idea of supervisor burden as the expected total cost incurred by the human in providing interventions, which consists of the expected cost due to *context switching* between autonomous and human control and the time spent actually providing interventions. However, it is difficult to design intervention criteria that limit this burden while ensuring that the robot gains sufficient information to imitate the supervisor's policy.

This paper makes several contributions. First, we develop intervention criteria based on a synthesis of two estimated properties of a given state: *novelty*, which measures whether the state is significantly out of the distribution of previously encountered states, indicating that the robot policy should not be trusted; and *risk*, which measures the likelihood of the robot successfully completing the task on its own. While state novelty has been considered in prior work [26], the key insight in our intervention criteria lies in combining novelty with a new risk metric to estimate the probability of task success. Second, we present a new robot-gated interactive IL algorithm, ThriftyDAgger (Figure 1), which employs these measures jointly to solicit human interventions only when necessary. Third, while prior robot-gated algorithms [19, 18] require careful parameter tuning to modulate the timing and frequency of human intervention requests, ThriftyDAgger only requires the supervisor to specify a desired context switching rate and sets thresholds accordingly. Fourth, experimental results demonstrate ThriftyDAgger's effectiveness for reducing supervisor burden while learning challenging tasks both in simulation and in an image-based cable routing task on a physical robot. Finally, the results of a human user study applying ThriftyDAgger to control a fleet of three simulated robots suggest that ThriftyDAgger significantly improves performance on both the robots' task and an independent human task while imposing fewer context switches, fewer human intervention actions, and lower mental load and frustration than prior algorithms.

## 2   Related Work

**Imitation Learning from Human Feedback:** There has been significant prior work in offline imitation learning, in which the agent leverages an offline dataset of expert demonstrations either to directly match the distribution of trajectories in the offline dataset [4, 27, 1, 3, 28, 29, 30], for instance via Behavior Cloning [31, 32], or to learn a reward function that can then be optimized via reinforcement learning [33, 27, 34]. However, while these approaches have shown significant success in a number of domains [7, 10, 9, 32], learning from purely offline data leads to a trajectory distribution mismatch which yields suboptimal performance both in theory and practice [12, 13]. To address this problem, there have been a number of approaches that utilize online human feedback while the agent acts in the environment, such as providing suggested actions [12, 35, 36, 17] or preferences [37, 38, 39, 40, 41, 42]. However, many of these forms of human feedback may be unreliable if the robot visits states that significantly differ from those the human supervisor would

themselves visit; in such situations, it is challenging for the supervisor to determine what correct behavior should look like without directly interacting with the environment [16, 43].

**Interactive Imitation Learning:** A natural way to collect reliable online feedback for imitation learning is to periodically cede control to a human supervisor, who then provides a corrective intervention to illustrate desired behavior. Human-gated interactive IL algorithms [15, 16, 24] such as HG-DAgger [15] require the human to determine when to engage in interventions. However, these algorithms require a human to continuously monitor the robot to determine when to intervene, which imposes significant burden on the supervisor and is particularly impractical if a small number of humans must supervise a large number of robots. Furthermore, it requires the human to determine when the robot needs help and when to cede control, which can be unintuitive and unreliable.

By contrast, robot-gated interactive IL algorithms, such as EnsembleDAgger [26], SafeDAgger [19], and LazyDAgger [18], allow the robot to actively query for human interventions. In practice, these algorithms estimate various quantities correlated with task performance [19, 18, 44, 25] and uncertainty [26] and use them to determine when to solicit interventions. Prior work has proposed intervention criteria which use the novelty of states visited by the robot [26] or the predicted discrepancy between the actions proposed by the robot policy and those of the supervisor [19, 18]. However, while state novelty provides a valuable signal for soliciting interventions, we argue that this alone is insufficient, as a state's novelty does not convey information about the level of precision with which actions must be executed in that state. In practice, many robotic tasks involve moving through critical "bottlenecks" [24], which, though not necessarily novel, still present challenges. Examples include moving an eating utensil close to a person's mouth or placing an object on a shelf without disturbing nearby objects. Similarly, even if predicted accurately, action discrepancy is often a flawed risk measure, as high action discrepancy between the robot and the supervisor may be permissible when fine-grained control is not necessary (e.g. a robot gripper moving in free space) but impermissible when precision is critical (e.g. a robot gripper actively trying to grasp an object). In contrast, ThriftyDAgger presents an intervention criteria incorporating both state novelty and a novel risk metric and automatically tunes key parameters, allowing efficient use of human supervision.

## 3  Problem Statement

Given a robot, a task for the robot to accomplish, and a human supervisor with a specified context switching budget, the goal is to train the robot to imitate supervisor performance within the budget. We model the robot environment as a discrete-time Markov Decision Process (MDP) $\mathcal{M}$ with continuous states $s \in \mathcal{S}$, continuous actions $a \in \mathcal{A}$, and time horizon $T$ [45]. We consider the interactive imitation learning (IL) setting [15], where the robot does not have access to a shaped reward function or to the MDP's transition dynamics but can temporarily cede control to a supervisor who uses policy $\pi_h : \mathcal{S} \to \mathcal{A}$. We specifically focus on tasks where there is a goal set $\mathcal{G}$ which determines success, but that can be challenging and long-horizon, making direct application of RL highly sample inefficient.

We assume that the human and robot utilize the same action space (e.g. through a teleoperation interface) and that task success can be specified by convergence to some goal set $\mathcal{G} \subseteq \mathcal{S}$ within the time horizon (i.e., the task is successful if $\mathcal{G}$ is reached within $T$ timesteps). We further assume access to an indicator function $\mathbb{1}_{\mathcal{G}} : \mathcal{S} \to \{0, 1\}$, which indicates whether a state belongs to the goal set $\mathcal{G}$.

The IL objective is to minimize a surrogate loss function $J(\pi_r)$ to encourage the robot policy $\pi_r : \mathcal{S} \to \mathcal{A}$ to match $\pi_h$:

$$J(\pi_r) = \sum_{t=1}^{T} \mathbb{E}_{s_t \sim d_t^{\pi_r}} \left[ \mathcal{L}(\pi_r(s_t), \pi_h(s_t)) \right], \tag{1}$$

where $\mathcal{L}(\pi_r(s), \pi_h(s))$ is an action discrepancy measure between $\pi_r(s)$ and $\pi_h(s)$ (e.g. MSE loss), and $d_t^{\pi_r}$ is the marginal state distribution at timestep $t$ induced by the robot policy $\pi_r$ in $\mathcal{M}$.

In the interactive IL setting, in addition to optimizing Equation (1), a key design goal is to minimize the imposed burden on the human supervisor. To formalize this, we define a switching policy $\pi$, which determines whether the system is under robot control $\pi_r$ (which we call *autonomous mode*) or human supervisor control $\pi_h$ (which we call *supervisor mode*). Following prior work [18], we define $C(\pi)$, the expected number of *context switches* in an episode under policy $\pi$, as follows: $C(\pi) = \sum_{t=1}^{T} \mathbb{E}_{s_t \sim d_t^{\pi}} [m_I(s_t; \pi)]$, where $m_I(s_t; \pi)$ is an indicator for whether or not a context switch occurs from autonomous to supervisor control. Similarly, we define $I(\pi)$ as the expected

number of *supervisor actions* in an intervention solicited by $\pi$. We then define the total burden $B(\pi)$ imposed on the human supervisor as follows:

$$B(\pi) = C(\pi) \cdot \big(L + I(\pi)\big), \qquad (2)$$

where $L$ is the *latency* of a context switch between control modes (summed over both switching directions) in units of timesteps, where each action takes one timestep. The interactive IL objective is to minimize the discrepancy from the supervisor policy while limiting supervisor burden within some $\Gamma_{\mathrm{b}}$:

$$\pi = \arg\min_{\pi' \in \Pi}\{J(\pi_r) \mid B(\pi') \leq \Gamma_{\mathrm{b}}\}. \qquad (3)$$

Because it is challenging to explicitly optimize policies to satisfy the supervisor burden constraint in Equation (3), we present novel intervention criteria that enable reduction of supervisor burden by limiting the total number of interventions to a user-specified budget. Given sufficiently high latency $L$, limiting the interventions $C(\pi)$ directly corresponds to limiting supervisor burden $B(\pi)$.

## 4 ThriftyDAgger

ThriftyDAgger determines when to switch between autonomous and human supervisor control modes by leveraging estimates of both the *novelty* and *risk* of states. Below, Sections 4.1 and 4.2 discuss the estimation of state novelty and risk of task failure, respectively, while Section 4.3 discusses ThriftyDAgger's integration of these measures to determine when to switch control modes. Section 4.4 then describes an online procedure to set thresholds for switching between control modes. Finally, Section 4.5 describes the full control flow of ThriftyDAgger.

### 4.1 Novelty Estimation

When the robot policy visits states that lie significantly outside the distribution of those encountered in the supervisor trajectories, it does not have any reference behavior to imitate. This motivates initiating interventions to illustrate desired recovery behaviors in these states. However, estimating the support of the state distribution visited by the human supervisor is challenging in the high-dimensional state spaces common in robotics. Following prior work [26], we train an ensemble of policies with bootstrapped samples of transitions from supervisor trajectories. We then measure the novelty of a given state $s$ by calculating the variance of the policy outputs at state $s$ across ensemble members. In practice, the action $a \in \mathcal{A}$ outputted by each policy is a vector; thus, we measure state novelty by computing the variance of each component of the action vector $a$ across the ensemble members and then averaging over the components. We denote this quantity by Novelty($s$). Once in supervisor mode, as noted in Hoque et al. [18], we can obtain a more precise correlate of novelty by computing the ground truth action discrepancy between the supervisor's actions and those of the robot policy.

### 4.2 Risk Estimation

Interventions may be required not only in novel states outside the distribution of supervisor trajectories, but also in familiar states that are prone to result in task failure. For example, a task might have a "bottleneck" region with low tolerance for error, which has low novelty but nevertheless requires more supervision to learn a reliable robot policy. To address this challenge, we propose a novel measure of a state's "riskiness," capturing the likelihood that the robot cannot successfully converge to the goal set $\mathcal{G}$. We first define a Q-function to quantify the discounted probability of successful convergence to $\mathcal{G}$ from a given state and action under the robot policy:

$$Q_{\mathcal{G}}^{\pi_r}(s_t, a_t) = \mathbb{E}_{s_{t'} \sim d_{t'}^{\pi_r}} \left[ \sum_{t'=t}^{\infty} \gamma^{t'-t} \mathbb{1}_{\mathcal{G}}(s_{t'}) | s_t, a_t \right], \qquad (4)$$

where $\mathbb{1}_{\mathcal{G}}(s_t)$ is equal to 1 if $s_t$ belongs to $\mathcal{G}$. We estimate $Q_{\mathcal{G}}^{\pi_r}(s_t, a_t)$ via a function approximator $\hat{Q}_{\phi,\mathcal{G}}^{\pi_r}$ parameterized by $\phi$, and define a state's riskiness in terms of this learned Q-function:

$$\mathrm{Risk}^{\pi_r}(s, a) = 1 - \hat{Q}_{\phi,\mathcal{G}}^{\pi_r}(s, a). \qquad (5)$$

In practice, we train $\hat{Q}_{\phi,\mathcal{G}}^{\pi_r}$ on transitions $(s_t, a_t, s_{t+1})$ from both autonomous mode and supervisor mode by minimizing the following MSE loss inspired by [46]:

$$J_{\mathcal{G}}^Q(s_t, a_t, s_{t+1}; \phi) = \frac{1}{2}\left(\hat{Q}_{\phi,\mathcal{G}}^{\pi_r}(s_t, a_t) - (\mathbb{1}_{\mathcal{G}}(s_t) + (1 - \mathbb{1}_{\mathcal{G}}(s_t))\gamma \hat{Q}_{\phi,\mathcal{G}}^{\pi_r}(s_{t+1}, \pi_r(s_{t+1})))\right)^2. \qquad (6)$$

Note that since $\hat{Q}_{\phi,\mathcal{G}}^{\pi_r}$ is only used to solicit interventions, it must only be accurate enough to distinguish risky states from others, rather than be able to make the fine-grained distinctions between different states required for accurate policy learning in reinforcement learning.

## 4.3 Regulating Switches in Control Modes

We now describe how ThriftyDAgger leverages the novelty estimator from Section 4.1 and the risk estimator from Section 4.2 to regulate switches between autonomous and supervisor control. While in autonomous mode, the switching policy $\pi$ initiates a switch to supervisor mode at timestep $t$ if either (1) state $s_t$ is sufficiently unfamiliar or (2) the robot policy has a low probability of task success from $s_t$. Stated precisely, $\pi$ initiates a switch to supervisor mode from autonomous mode at timestep $t$ if the predicate Intervene$(s_t, \delta_h, \beta_h)$ evaluates to TRUE, where Intervene$(s_t, \delta_h, \beta_h)$ is TRUE if (1) Novelty$(s_t) > \delta_h$ or (2) Risk$^{\pi_r}(s_t, \pi_r(s_t)) > \beta_h$, and FALSE otherwise. Note that the proposed switching policy only depends on Risk$^{\pi_r}$ for states which are *not* novel (as novel states already initiate switches to supervisor control regardless of risk), since the learned risk measure should only be trusted on states in the neighborhood of those on which it has been trained.

In supervisor mode, $\pi$ switches to autonomous mode if the action discrepancy between the human and robot policy and the robot's task failure risk are both below threshold values (Section 4.4), indicating that the robot is in a familiar and safe region. Stated precisely, $\pi$ switches to autonomous mode from supervisor mode if the predicate Cede$(s_t, \delta_r, \beta_r)$ evaluates to TRUE, where Cede$(s_t, \delta_r, \beta_r)$ is TRUE if (1) $||\pi_r(s_t) - \pi_h(s_t)||_2^2 < \delta_r$ and (2) Risk$^{\pi_r}(s_t, \pi_r(s_t)) < \beta_r$, and FALSE otherwise. Here, the risk metric ensures that the robot has a high probability of autonomously completing the task, while the coarser 1-step action discrepancy metric verifies that we are in a familiar region of the state space where the $\hat{Q}^{\pi_r}_{\phi,\mathcal{G}}$ values can be trusted. Motivated by prior work [18] and hysteresis control [47], we use asymmetric switching criteria with stricter thresholds in supervisor mode ($\beta_r < \beta_h$) to encourage lengthier interventions and reduce context switches experienced by the human supervisor.

## 4.4 Computing Risk and Novelty Thresholds from Data

One challenge of the control strategy presented in Section 4.3 lies in tuning the key parameters $(\delta_h, \delta_r, \beta_h, \beta_r)$ governing when context switching occurs. As noted in prior work [26], performance and supervisor burden can be sensitive to these thresholds. To address this difficulty, we assume that the user specifies their availability in the form of a desired intervention budget $\alpha_h \in [0, 1]$, indicating the desired proportion of timesteps in which interventions will be requested. This desired context switching rate can be interpreted in the context of supervisor burden as defined in Equation (2): if the latency of a context switch dominates the time cost of the intervention itself, limiting the expected number of context switches to within some intervention budget directly limits supervisor burden.

Given $\alpha_h$, we set $\beta_h$ to be the $(1 - \alpha_h)$-quantile of Risk$^{\pi_r}(s, \pi_r(s))$ for all states previously visited by $\pi_r$ and set $\delta_h$ to be the $(1 - \alpha_h)$-quantile of Novelty$(s)$ for all states previously visited by $\pi_r$. We set $\delta_r$ to be the mean action discrepancy on the states visited by the supervisor after $\pi_r$ is trained and set $\beta_r$ to be the median of Risk$^{\pi_r}(s, \pi_r(s))$ for all states previously visited by $\pi_r$. (Note that $\beta_r$ can easily be set to different quantiles to adjust mean intervention length if desired.) We find that these settings strike a balance between informative interventions and imposed supervisor burden.

## 4.5 ThriftyDAgger Overview

We now summarize the ThriftyDAgger procedure, with full pseudocode available in the supplement. ThriftyDAgger first initializes $\pi_r$ via Behavior Cloning on offline transitions ($\mathcal{D}_h$ from the human supervisor, $\pi_h$). Then, $\pi_r$ collects an initial offline dataset $\mathcal{D}_r$ from the resulting $\pi_r$, initializes $\hat{Q}^{\pi_r}_{\phi,\mathcal{G}}$ by optimizing Equation (5) on $\mathcal{D}_r \cup \mathcal{D}_h$, and initializes parameters $\beta_h, \beta_r, \delta_h$, and $\delta_r$ as in Section 4.4. We then collect data for $N$ episodes, each with up to $T$ timesteps. In each timestep of each episode, we determine whether robot policy $\pi_r$ or human supervisor $\pi_h$ should be in control using the procedure in Section 4.3. Transitions in autonomous mode are aggregated into $\mathcal{D}_r$ while transitions in supervisor mode are aggregated into $\mathcal{D}_h$. After each episode, $\pi_r$ is updated via supervised learning on $\mathcal{D}_h$, and $\hat{Q}^{\pi_r}_{\phi,\mathcal{G}}$ is then updated on $\mathcal{D}_r \cup \mathcal{D}_h$ to reflect the probability of task success of the updated $\pi_r$.

# 5 Experiments

In the following experiments, we study whether ThriftyDAgger can balance task performance and supervisor burden more effectively than prior IL algorithms in three contexts: (1) training a simulated robot to perform a peg insertion task (Section 5.3); (2) supervising a fleet of three simulated robots to perform the peg insertion task in a human user study (Section 5.4); and (3) training a physical surgical robot to perform a cable routing task (Section 5.5). In the supplementary material, we also include results from an additional simulation experiment on a challenging block stacking task.

## 5.1 Evaluation Metrics

We consider ThriftyDAgger's performance during training and execution. For the latter, we evaluate both the (1) *autonomous success rate*, or success rate when deployed after training without access to a human supervisor, and (2) *intervention-aided success rate*, or success rate when deployed after training with a human supervisor in the loop. These metrics are reported in the Peg Insertion study (Section 5.3) and the Physical Cable Routing study (Section 5.5). For all experiments, during both training and intervention-aided execution, we evaluate the number of interventions, human actions, and robot actions per episode. These metrics are computed over successful episodes only to prevent biasing the metrics by the maximum episode horizon length $T$; such bias occurs, for instance, when less successful policies appear to take more actions due to hitting the time boundary more often. Additional metrics including cumulative statistics across all episodes are reported in the supplement. In our user study (Section 5.4), we also report the following quantities: throughput (total number of task successes across the three robots), performance on an independent human task, the idle time of the robots in the fleet, and users' qualitative ratings of mental load and frustration. By comparing the amount of human supervision and success rates across different algorithms, we are interested in evaluating how effectively each algorithm balances supervision with policy performance.

## 5.2 Comparisons

We compare ThriftyDAgger to the following algorithms: Behavior Cloning, which does not use interventions; HG-DAgger [15], which is human-gated and always requires supervision; SafeDAgger [19], which is robot-gated and performs interventions based on estimated action discrepancy between the human supervisor and robot policy; and LazyDAgger [18], which builds on SafeDAgger by introducing asymmetric switching criteria to encourage lengthier interventions. We also implement two ablations: one that does not use a novelty measure to regulate context switches (ThriftyDAgger (-Novelty)) and one that does not use risk to regulate context switches (ThriftyDAgger (-Risk)).

## 5.3 Peg Insertion in Simulation

We first evaluate ThriftyDAgger on a long-horizon (100+ timesteps) peg insertion task (Figure 2) from the Robosuite simulation environment [48]. The goal is to grasp a ring in a random initial pose and thread it over a cylinder at a fixed target location. This task has two bottlenecks which motivate learning from interventions: (1) correctly grasping the ring and (2) correctly placing it over the cylinder. A human teleoperates the robot through a keyboard interface to provide interventions. The states consist of the robot's joint angles and ring's pose, while the actions specify 3D translation, rotation, and opening or closing the gripper. For ThriftyDAgger and its ablations, we use target intervention frequency $\alpha_h = 0.01$ (Section 4.4). We collect 30 offline task demos (2,687 state-action pairs) from a human supervisor to initialize the robot policy for all compared algorithms. Behavior Cloning is given additional state-action pairs roughly equivalent to the average amount of supervisor actions solicited by the interactive algorithms (Table 4 in the appendix). For ThriftyDAgger and each interactive IL baseline, we perform 10,000 environment steps, during which each episode takes at most 175 timesteps and system control switches between the human and robot. Hyperparameter settings for all algorithms are detailed in the supplement.

Results (Table 1) suggest that ThriftyDAgger simultaneously solicits fewer interventions and achieves a significantly higher autonomous success rate than prior robot-gated algorithms, although it does request more human actions due to its conservative exit criterion for interventions (Cede$(s_t, \delta_r, \beta_r)$). The number of human actions falls significantly at execution time (Table 1), when the robot policy has been trained on online data and is therefore less risky. We find that all interactive IL algorithms substantially outperform Behavior Cloning, which does not have access to supervisor interventions. Notably, ThriftyDAgger achieves a higher autonomous success rate than even HG-DAgger, in which the supervisor is able to decide the timing and length of interventions. This indicates that ThriftyDAgger's intervention criteria enable it to autonomously solicit interventions as informative as those chosen by a human supervisor with expert knowledge of the task. Furthermore, ThriftyDAgger achieves a 100% intervention-aided success rate at execution time, suggesting that ThriftyDAgger successfully identifies the required states at which to solicit interventions. We find that both ablations of ThriftyDAgger (Ours (-Novelty) and Ours (-Risk)) achieve significantly lower autonomous success rates, indicating that both the novelty and risk measures are critical to ThriftyDAgger's performance. We calculate ThriftyDAgger's context switching rate to be 1.15% novelty switches and 0.79% risk switches, both approximately within the budget of $\alpha_h = 0.01$.

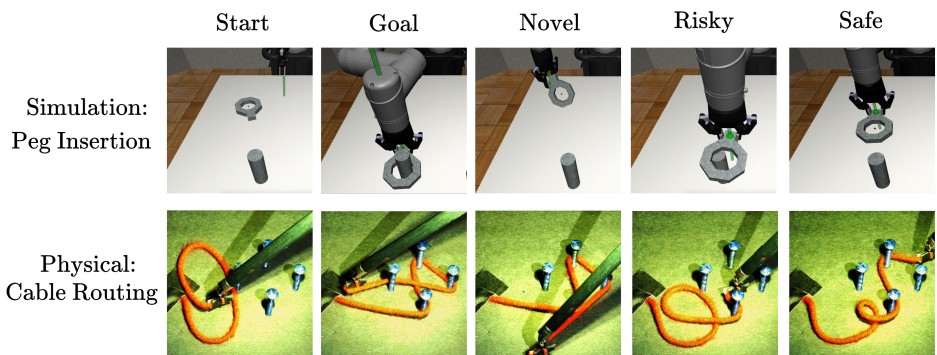

|  | Start | Goal | Novel | Risky | Safe |
|--|-------|------|-------|-------|------|
| Simulation: Peg Insertion | | | | | |
| Physical: Cable Routing | | | | | |

Figure 2: **Experimental Domains:** We visualize the peg insertion simulation domain (top row) and the cable routing domain with the physical robot (bottom row). We visualize sample start and goal states, in addition to states which ThriftyDAgger categorizes as novel, risky, or neither. ThriftyDAgger marks states as novel if they are far from states that the supervisor visited and risky if the robot is stuck in a bottleneck, e.g. if the ring is wedged against the side of the cylinder (top) or the cable is near all four obstacles (bottom).

Table 1: **Peg Insertion in Simulation Results:** We first report training performance (number of interventions (Ints), number of human actions (Acts (H)), and number of robot actions (Acts (R))) and report the success rate of the fully-trained policy at execution time when no interventions are allowed (Auto Succ.). We then evaluate the fully-trained policies with interventions allowed and report the same intervention statistics and the success rate (Int-Aided Succ.). We find that ThriftyDAgger achieves the highest autonomous and intervention-aided success rates among all algorithms compared. Notably, ThriftyDAgger even achieves a higher autonomous success rate than HG-DAgger, in which the human decides when to intervene during training.

| Algorithm | Training Interventions | | | Auto Succ. | Execution Interventions | | | Int-Aided Succ. |
|-----------|------|--------|--------|------------|------|--------|--------|----------------|
|           | Ints | Acts (H) | Acts (R) | | Ints | Acts (H) | Acts (R) | |
| Behavior Cloning | N/A | N/A | $108.0 \pm 15.9$ | $24/100$ | N/A | N/A | N/A | N/A |
| SafeDAgger | $3.89 \pm 1.44$ | $19.8 \pm 9.9$ | $88.8 \pm 19.4$ | $24/100$ | $4.00 \pm 1.37$ | $19.5 \pm 5.3$ | $77.5 \pm 11.7$ | $17/20$ |
| LazyDAgger | $1.46 \pm 1.15$ | $13.2 \pm 12.4$ | $102.1 \pm 18.2$ | $48/100$ | $1.73 \pm 1.29$ | $12.6 \pm 14.4$ | $91.7 \pm 24.0$ | $11/20$ |
| HG-DAgger | $1.49 \pm 0.88$ | $20.3 \pm 15.6$ | $97.1 \pm 17.5$ | $57/100$ | $1.15 \pm 0.73$ | $17.1 \pm 11.6$ | $103.6 \pm 14.0$ | $\mathbf{20/20}$ |
| Ours (-Novelty) | $\mathbf{0.79 \pm 0.81}$ | $35.1 \pm 23.1$ | $70.0 \pm 35.8$ | $49/100$ | $\mathbf{0.33 \pm 0.62}$ | $2.5 \pm 5.0$ | $114.0 \pm 26.0$ | $12/20$ |
| Ours (-Risk) | $0.99 \pm 0.96$ | $7.8 \pm 12.0$ | $104.2 \pm 19.2$ | $49/100$ | $1.39 \pm 0.95$ | $9.8 \pm 12.0$ | $109.1 \pm 22.9$ | $18/20$ |
| Ours: ThriftyDAgger | $0.88 \pm 1.01$ | $43.6 \pm 24.5$ | $60.0 \pm 32.8$ | $\mathbf{73/100}$ | $1.35 \pm 0.66$ | $21.3 \pm 15.0$ | $84.8 \pm 21.8$ | $\mathbf{20/20}$ |

## 5.4 User Study: Controlling A Fleet of Three Robots in Simulation

We conduct a user study with 10 participants (7 male and 3 female, aged 18-37). Participants supervise a fleet of three simulated robots, each performing the peg insertion task from Section 5.3. We evaluate how different interactive IL algorithms affect the participants' (1) ability to provide effective robot interventions, (2) performance on a distractor task performed between robot interventions, and (3) levels of mental demand and frustration. For the distractor task, we use the game Concentration (also known as Memory or Matching Pairs), in which participants identify as many pairs of matching cards as possible among a set of face-down cards. This is intended to emulate tasks which require continual focus, such as cooking a meal or writing a research paper, in which frequent context switches between performing the task and helping the robots is frustrating and degrades performance.

The participants teleoperate the robots using three robot-gated interactive IL algorithms: SafeDAgger, LazyDAgger, and ThriftyDAgger. The participant is instructed to make progress on the distractor task only when no robot requests an intervention. When an intervention is requested, the participant is instructed to pause the distractor task, provide an intervention from the requested state until the robot (or multiple robots queued after each other) no longer requires assistance, and then return to the distractor task. The participants also teleoperate with HG-DAgger, where they no longer perform the distractor task and are instructed to continually monitor all three robots simultaneously and decide on the length and timing of interventions themselves. Each algorithm runs for 350 timesteps, where in each timestep, all robots in autonomous mode execute one action and the human executes one action on the currently supervised robot (if applicable). The supplement illustrates the user study interface and fully details the experiment protocol. All algorithms are initialized as in Section 5.3.

Results (Table 2) suggest that ThriftyDAgger achieves significantly higher throughput than all prior algorithms while requiring fewer interventions and fewer human actions, indicating that ThriftyDAgger requests interventions more judiciously than prior algorithms. Furthermore, ThriftyDAgger also enables a lower mean idle time for robots and higher performance on the distractor task. Notably, ThriftyDAgger solicits fewer interventions and total actions while achieving a higher throughput than HG-DAgger, in which the participant chooses when to intervene. We also report metrics of users'

Table 2: **Three-Robot Fleet Control User Study Results:** Results for experiments with 10 human subjects and 3 simulated robots on the peg insertion task. We report the total numbers of interventions, human actions, and robot actions, as well as the throughput, or total task successes achieved across robots, for all algorithms. Additionally, for robot-gated algorithms, we report the Concentration score (number of pairs found) and the mean idle time of robots in the fleet in timesteps. Results suggest that ThriftyDAgger outperforms all prior algorithms across all metrics, requesting fewer interventions and total human actions while achieving higher throughput, lowering the robots' mean idle time, and enabling higher performance on the Concentration task.

| Algorithm | Interventions | Human Actions | Robot Actions | Concentration Pairs | Throughput | Mean Idle Time |
|---|---|---|---|---|---|---|
| HG-DAgger | $10.6 \pm 2.5$ | $198.0 \pm 32.1$ | $834.4 \pm 38.1$ | N/A | $5.1 \pm 1.9$ | N/A |
| SafeDAgger | $22.1 \pm 4.8$ | $234.1 \pm 31.8$ | $700.7 \pm 70.4$ | $17.7 \pm 8.2$ | $3.0 \pm 2.4$ | $38.4 \pm 14.1$ |
| LazyDAgger | $10.0 \pm 2.1$ | $219.5 \pm 43.3$ | $719.2 \pm 89.7$ | $20.9 \pm 7.9$ | $5.1 \pm 1.7$ | $37.1 \pm 20.5$ |
| Ours: ThriftyDAgger | $\mathbf{7.9 \pm 2.1}$ | $\mathbf{179.4 \pm 34.9}$ | $793.2 \pm 86.6$ | $\mathbf{33.0 \pm 8.5}$ | $\mathbf{9.2 \pm 2.0}$ | $\mathbf{25.8 \pm 19.3}$ |

Table 3: **Physical Cable Routing Results:** We first report intervention statistics during training (number of interventions (Ints), number of human actions (Acts (H)), and number of robot actions (Acts (R))) and report the success rate of the fully-trained policy at execution time when no interventions are allowed (Auto Succ.). We then evaluate the fully-trained policies with interventions allowed and report the same intervention statistics and the success rate (Int-Aided Succ.). We find that ThriftyDAgger achieves the highest autonomous and intervention-aided success rates among all algorithms compared. Notably, ThriftyDAgger achieves a comparable autonomous success rate to HG-DAgger, in which the human decides when to intervene during training.

| Algorithm | Training Interventions | | | Auto Succ. | Execution Interventions | | | Int-Aided Succ. |
|---|---|---|---|---|---|---|---|---|
| | Ints | Acts (H) | Acts (R) | | Ints | Acts (H) | Acts (R) | |
| Behavior Cloning | N/A | N/A | N/A | 0/15 | N/A | N/A | N/A | N/A |
| HG-DAgger | $1.55 \pm 1.16$ | $13.9 \pm 10.9$ | $55.5 \pm 10.9$ | 10/15 | $\mathbf{0.40 \pm 0.49}$ | $2.7 \pm 3.5$ | $73.9 \pm 7.9$ | **15/15** |
| Ours: ThriftyDAgger | $\mathbf{1.42 \pm 1.14}$ | $15.2 \pm 12.4$ | $45.5 \pm 18.3$ | **12/15** | $0.40 \pm 0.71$ | $1.5 \pm 3.1$ | $61.3 \pm 6.5$ | **15/15** |

mental workload and frustration using the NASA-TLX scale [49] in the supplement. Results suggest that users experience lower degrees of frustration and mental load when interacting with ThriftyDAgger and LazyDAgger compared to HG-DAgger and SafeDAgger. We hypothesize that participants struggle with HG-DAgger due to the difficultly of monitoring multiple robots simultaneously, while SafeDAgger's frequent context switches lead to user frustration during experiments.

## 5.5 Physical Experiment: Visuomotor Cable Routing

Finally, we evaluate ThriftyDAgger on a long-horizon cable routing task with a da Vinci surgical robot [50]. Here, the objective is to route a red cable into a Figure-8 pattern around 4 pegs via teleoperation with the robot's master controllers (see supplement). The algorithm only observes high-dimensional $64 \times 64 \times 3$ RGB images of the workspace and generates continuous actions representing delta-positions in $(x, y)$. As in Section 5.3, ThriftyDAgger uses a target intervention frequency of $\alpha_h = 0.01$. We collect 25 offline task demonstrations (1,381 state-action pairs) from a human supervisor to initialize the robot policy for ThriftyDAgger and all comparisons. We perform 1,500 environment steps, where each episode has at most 100 timesteps and system control can switch between the human and robot. The supplement details the hyperparameter settings for all algorithms.

Results (Table 3) suggest that both ThriftyDAgger and HG-DAgger achieve a significantly higher autonomous success rate than Behavior Cloning, which is never able to complete the task. Furthermore, ThriftyDAgger achieves a comparable autonomous success rate to HG-DAgger while requesting fewer interventions and a similar number of total human actions. This again suggests that ThriftyDAgger's intervention criteria enable it to solicit interventions equally as informative or more informative than those chosen by a human supervisor. Finally, at execution time ThriftyDAgger achieves a 100% intervention-aided success rate with minimal supervision, again indicating that ThriftyDAgger successfully identifies the timing and length of interventions to increase policy reliability.

## 6 Discussion and Future Work

We present ThriftyDAgger, a scalable robot-gated interactive imitation learning algorithm that leverages learned estimates of state novelty and risk of task failure to reduce burden on a human supervisor during training and execution. Experiments suggest that ThriftyDAgger effectively enables long-horizon robotic manipulation tasks in simulation, on a physical robot, and for a three-robot fleet while limiting burden on a human supervisor. In future work, we hope to apply ideas from ThriftyDAgger to interactive reinforcement learning and larger scale fleets of physical robots. We also hope to study how ThriftyDAgger's performance varies with the target supervisor burden specified via $\alpha_h$. In practice, $\alpha_h$ could even be time-varying: for instance, $\alpha_h$ may be significantly lower at night, when human operators may have limited availability. Similarly, $\alpha_h$ may be set to a higher value during training than at deployment, when the robot policy is typically higher quality.

**Acknowledgments**

This research was performed at the AUTOLAB at UC Berkeley in affiliation with the Berkeley AI Research (BAIR) Lab and the CITRIS "People and Robots" (CPAR) Initiative. The authors were supported in part by the Scalable Collaborative Human-Robot Learning (SCHooL) Project, NSF National Robotics Initiative Award 1734633, and by donations from Google, Siemens, Amazon Robotics, Toyota Research Institute, Autodesk, Honda, Intel, and Hewlett-Packard and by equipment grants from PhotoNeo, NVidia, and Intuitive Surgical. Any opinions, findings, and conclusions or recommendations expressed in this material are those of the author(s) and do not necessarily reflect the views of the sponsors. We thank our colleagues who provided helpful feedback, code, and suggestions, especially Anca Dragan, Vincent Lim, and Zaynah Javed.

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
