# OpenReview forum: "ThriftyDAgger: Budget-Aware Novelty and Risk Gating for Interactive Imitation Learning"
_robot-learning.org/CoRL/2021/Conference — CoRL2021 Oral_

### Official Review · Reviewer_PRtc · 2021-07-22

**Originality:** Good
**Technical Quality:** Good
**Clarity Of Presentation:** Very Good
**Impact:** 3

**Recommendation:**

Weak Accept: I recommend accepting the paper, but will not argue for my recommendation if the majority of other reviewers have a different opinion.

**Summary:**

This paper addresses the problem of robot-gated interactive imitation learning, where a robot actively cedes control to a human operator in situations where it requires action supervision. Compare to prior work, the primary novelty is the inclusion of a "risk" term (in addition to a "novelty" term) to determine when the robot should cede control to the human operator, or take control back from the human operator. The method is shown to outperform baselines in terms of success rate and amount of human effort needed on a peg insertion task in simulation and a cable routing task on a physical robot. A user study is also conducted, where users attempt to focus on a pattern matching game, while also trying to provide interventions (as requested) by a fleet of 3 robot arms that are trying to learn the peg insertion task in simulation.

**Issues:**

Discuss the role of the risk term - why it is expected to help significantly, especially in the context of prior work that does not consider it (like LazyDagger and SafeDagger).

**Reviewer Expertise:**

Very good: Comprehensive knowledge of the area

**Strengths And Weaknesses:**

Strengths

The paper is well-written and tackles an important problem. The proposed method is shown to compare favorably to several baselines on a simulated peg insertion task and a real cable routing task, in terms of the amount of human effort needed to learn the task, and the final performance of the trained model. The included user study across 10 users that each try to assist a fleet of 3 robot arms is also very interesting as a proof-of-concept for the proposed method. The parameters of the switching policy (which are usually manually specified in prior work) are determined from a single hyperparameter that correlates to the budget for the expected amount of effort a human is willing to provide in training the robot -- this is a significant improvement over other methods. The attached video also provides a useful overview.

Weaknesses

The two main weaknesses are the lack of novelty and breadth of the experiments. The main difference from LazyDagger appears to be the inclusion of a risk term, and automatic hyperparameter tuning. That being said, the risk term appears to empirically make a large difference, but there isn't as much intuition or justification provided in the paper for why it makes such a large difference. Furthermore, it's unclear why the proposed risk term works well - the risk term should vary naturally during the episode, by state. At the beginning of the episode, there are several timesteps left to reach the goal, and towards the end, there will be few. How can this term distinguish between states that are far away from the goal, and states that will never reach the goal? Providing some concrete qualitative examples of states along with the value of the risk term at those states might be useful to understand this.

Also, while the number of baselines is high, the number of tasks used for evaluation is lacking - only one task in simulation and one in real is used for evaluation. It would be useful to demonstrate the effectiveness of the method on at least one more simulated task, and preferably one that is more challenging than the peg insertion task (the fitting appears to be relatively high tolerance).

Additional comments follow:
- Reporting additional metrics could be beneficial to provide a more complete picture, with respect to the comparison to BC. Table 1 would ideally report the number of episodes / total number of samples used for training each algorithm (both human and robot samples). Including information on the number of successful and unsuccessful trajectories used for each method would be useful as well (perhaps in the supplementary), since the proposed method is able to leverage unsuccessful trajectories to train the risk metric (value function). It might also be useful to report other metrics like wall clock time - there's likely a tradeoff here (for example, perhaps 100 demonstrations could be collected in 1 hour of human time, allowing the robot to match the success rate of the proposed method, which might use 15 minutes of total human time, spread out over 3 hours. The proposed method would take more overall human + robot wall clock time, but less human time).
- The paper concludes that the proposed method outperforms HG-Dagger, after presenting the results in Table 1 on the peg insertion task. However, as mentioned previously, there should be at least 1 more task in simulation, to make the claim more solid, as the performance gap appears to be much tighter on the real world task. Furthermore, while the proposed algorithm outperforms HG-Dagger in the user study, the setup there was already biased against HG-Dagger, since participants must actively intervent on 3 robots operating at once (which is definitely cognitively demanding). Including further results in the single robot case across multiple users would also substantiate the claim on outperforming HG-Dagger further.
- From the supplementary material, it seems like every action in the peg insertion task is followed by 10 zero actions, in order "to avoid bias due to variable teleoperation speeds". This sounds like an unconventional design choice that could impact the difficulty of the task and could merit further discussion and motivation. This might also explain why the motion of the robot appears stilted in the included video (although that could also be an artifact of the teleoperation interface).
- What happens if the algorithm cedes control to the human, but the robot is in a state in which it is difficult or impossible to recover? Does the human just choose to terminate the episode? How often does this occur in practice?
- Line 328 - typo - "at which solicit interventions"

**Summary Of Recommendation:**

While the proposed method has some favorable characteristics and is shown to outperform baselines, the method is similar to prior work except for what appears to be one critical component (using the risk metric in determining when to switch between robot and human). Further discussion of this critical component would be beneficial. The breadth of the empirical evaluation is also limited.

---

> ### Author Response · Authors · 2021-08-21
> **Response to Reviewer PRtc**
>
> **Thank you for your insightful and helpful comments. We have addressed your comments below and uploaded a revision of the paper and supplement with all the changes indicated in the responses below highlighted in blue. Due to space constraints, our reply is split into two parts.**
>
> “That being said, the risk term appears to empirically make a large difference, but there isn't as much intuition or justification provided in the paper for why it makes such a large difference.”
>
> **We have revised the paper to elaborate on this point (Sections 4.2 and 4.3).  Our intuition is as follows. A robot trained with imitation learning should request help from a human in two scenarios: (1) when it has left the support of the expert’s data distribution or (2) when its policy will fail at the task. These conditions are not mutually exclusive, as a robot that has left the expert’s data distribution will likely fail at the task. Novelty estimation can capture Case 1 but not Case 2. Our insight is that the Case 2 states that do not fall under Case 1 must be familiar states; hence we can train and rely on a Q-function to accurately estimate the probability of task success. For example, “bottleneck regions” in the task (such as aligning the gripper with the peg in peg insertion) can require relatively precise manipulation, causing the robot policy to require more expert samples in this region than the amount of samples that results in a low novelty estimate. A more pathological case would be an inability of the policy class to capture the appropriate action, in which case the robot should always cede to the human in the dangerous region in order to succeed at the task at execution time. Furthermore, the robot should only take control back from the human when it is confident it can complete the task by itself. We leverage the risk metric here as well, in conjunction with a coarser 1-step action discrepancy to ensure we are in a familiar region and can therefore trust the Q-function.**
>
> “How can this [risk] term distinguish between states that are far away from the goal, and states that will never reach the goal?”
>
> **This is a great question. The risk term measures the discounted probability of reaching the goal in the future, so with a sufficiently high discount factor (we use gamma=0.9999), the risk term should still be able to distinguish between states from which the agent is likely to reach the goal, but only after many timesteps, and states from which the agent will never reach the goal. However, with a low discount factor, we agree that distinguishing between states very far away from the goal and states that will never reach the goal is challenging.**
>
> “It would be useful to demonstrate the effectiveness of the method on at least one more simulated task, and preferably one that is more challenging than the peg insertion task (the fitting appears to be relatively high tolerance).”
>
> **As suggested, we added results on a new block stacking task from the Robosuite simulation environment (see Section 7.3.3). This task is more difficult due to the randomized target position, lower tolerance placement region, and geometric symmetries in the blocks, as is evidenced by the lower autonomous success rates across all algorithms. As in the other environments, results suggest that ThriftyDAgger is competitive with HG-DAgger and outperforms other baselines and ablations in terms of success rate and number of interventions.**
>
> “Reporting additional metrics could be beneficial to provide a more complete picture, with respect to the comparison to BC. Table 1 would ideally report the number of episodes / total number of samples used for training each algorithm (both human and robot samples). Including information on the number of successful and unsuccessful trajectories used for each method would be useful as well (perhaps in the supplementary), since the proposed method is able to leverage unsuccessful trajectories to train the risk metric (value function). It might also be useful to report other metrics like wall clock time.”
>
> **Thank you for the suggestion. We added all these requested metrics to the appendix in Tables 4-7; note that we only had wall clock time metrics recorded for the user study (Table 7). We refer to the ratio of successful trajectories to all trajectories during training as the Train Success Rate, which gives insight into both the performance of the hybrid policy during training and the achieved throughput (by comparing the raw number of successes, as all algorithms are executed for the same number of time steps). Results suggest that ThriftyDAgger outperforms baselines and ablations and is competitive with HG-DAgger in terms of training success rate and throughput, but can request more total human samples than LazyDAgger and SafeDAgger due to the longer interventions.**

---

> ### Author Response · Authors · 2021-08-21
> **Response to Reviewer PRtc (Part 2)**
>
> “The paper concludes that the proposed method outperforms HG-Dagger, after presenting the results in Table 1 on the peg insertion task. However, as mentioned previously, there should be at least 1 more task in simulation to make the claim more solid, as the performance gap appears to be much tighter on the real world task. Furthermore, while the proposed algorithm outperforms HG-Dagger in the user study, the setup there was already biased against HG-Dagger, since participants must actively intervene on 3 robots operating at once (which is definitely cognitively demanding).”
>
> **In the revised version, we de-emphasized our claims that we outperform HG-DAgger in Section 5 and instead claim that we match its performance. This claim is further substantiated by the new block stacking simulation domain (Section 7.3.3). We believe that in the 1 robot case (as in both the simulation experiments without the user study and the physical experiment), even performing comparably to HG-DAgger is a surprising and compelling result, since this indicates that the intervention criteria learned by ThriftyDAgger are competitive with a human expert’s judgment on when to intervene.**
>
> “From the supplementary material, it seems like every action in the peg insertion task is followed by 10 zero actions, in order ‘to avoid bias due to variable teleoperation speeds’. This sounds like an unconventional design choice that could impact the difficulty of the task and could merit further discussion and motivation.”
>
> **We revised the paper to clarify this point (Section 7.3.1). By default, the Robosuite simulator records keyboard input at a fixed control frequency. This requires the human to be consistent in the timing of the demonstrations. For instance, if the human waits a bit too long, the keyboard will record no input, and the agent will incorrectly interpret the expert action at the current state as a no-op. Furthermore, previous actions can influence the future states due to momentum in the robot. To ensure the Markov property applies and avoid these erroneous action labels, for each action we add 10 no-ops (effectively letting the arm “settle”) and then pause the simulation until the human has entered another command through the keyboard. In practice, this does not make the task more difficult as the actions are still fine-grained enough for precise manipulation.**
>
> “What happens if the algorithm cedes control to the human, but the robot is in a state in which it is difficult or impossible to recover? Does the human just choose to terminate the episode? How often does this occur in practice?”
>
> **This is a great question. In this case, the human will act as well as they can but may not succeed at the task. The human is not allowed to terminate the episode prematurely; often an intervention solicited here will continue until the time boundary is hit, at which point the episode terminates as a failure. This occurs about 10% of the time in the peg insertion task (accounting for most of ThriftyDAgger’s failures during training time in Table 4), specifically when the washer is dropped prematurely and lands on its side near the target peg, making it very difficult to correct the orientation. Ideally the risk metric learns to anticipate these dangerous regions, and indeed we observe a higher success rate at execution time.**

---

> > ### Comment · Reviewer_PRtc · 2021-08-23
> > **Response**
> >
> > Thank you for your comments. I have further questions and comments below.
> >
> > Regarding the user interface, my worry was that waiting 10 timesteps between each input and pausing the simulation actually makes the task easier, not harder, as the robot does not need to deal with continuous movement. This is also a consideration for high-precision real-world tasks such as insertion, where it could be much harder (or impossible) for a human supervisor to perform low-tolerance insertion if not given continuous control of the system.
> >
> > I'd also like further clarification on the comparison with behavioral cloning - on line 263 of the current manuscript it is reported that 10000 env steps are used for each interactive learning algorithm, while it seems that 45 demonstrations are used for behavioral cloning. Please comment on why this is a fair comparison (e.g. is it because the total number of samples used for training the policy with the interactive learning methods is close to 15 demonstrations? how are the rest of the 10000 samples used?).
> >
> > I appreciate that the authors added a second simulated task, but the choice of stacking is a little unclear to me -- stacking seems like an easier task than peg insertion. Grasping the small box does not seem particularly difficult, and the precision needed to place the box on top of the large box seems pretty low as well. The low success rate of 5% for BC on 30 demonstrations also seems lower than expected - is there any intuition for why this might be the case?
> >
> > Finally, the response mentioned that humans are not allowed to terminate the episode early - can you clarify why this is the case? It seems like a strange choice. Does this also mean that all episodes during data collection are the same in length?

---

> > > ### Author Response · Authors · 2021-08-24
> > > **Response to Response (to Response)**
> > >
> > > Thank you Reviewer PRTc. Our response is split into 2 comments.
> > >
> > > Regarding the user interface, my worry was that waiting 10 timesteps between each input and pausing the simulation actually makes the task easier, not harder, as the robot does not need to deal with continuous movement. This is also a consideration for high-precision real-world tasks such as insertion, where it could be much harder (or impossible) for a human supervisor to perform low-tolerance insertion if not given continuous control of the system.
> > >
> > > **This is a good point. Our design choice here was to avoid polluting the demonstration data with variable teleoperation speed and unintentional no-ops at the cost of making the task somewhat easier. However we still believe the tasks are sufficiently difficult (see the third response), as we identified where BC falls short and developed an algorithm to address this. Furthermore, the real-world experiment in the paper does allow continuous movement: we set the control frequency at 1 Hz without pausing between supervisor actions. For extremely precise tasks such as needle insertion, we could increase this control frequency appropriately. When this control frequency becomes very high, one option is to have a RNN-based policy that outputs an action based on a sequence of past states instead of only the current state.**
> > >
> > > I'd also like further clarification on the comparison with behavioral cloning - on line 263 of the current manuscript it is reported that 10000 env steps are used for each interactive learning algorithm, while it seems that 45 demonstrations are used for behavioral cloning. Please comment on why this is a fair comparison (e.g. is it because the total number of samples used for training the policy with the interactive learning methods is close to 15 demonstrations? how are the rest of the 10000 samples used?).
> > >
> > > **Thanks for pointing this out; we revised Section 5.3 in the manuscript to add clarification. Our goal was to make the amount of expert action labels comparable across the algorithms. From Table 4 in the supplementary material we see that on average each interactive imitation learning algorithm (all algorithms other than behavior cloning) end up using about 4600 supervisor actions in total (offline + online). As a result, to make the comparison more fair, we supplied behavior cloning with additional offline demonstrations so that it received 4004 total supervisor actions. While online data may be qualitatively different from offline data, this is not something we can control for in an offline algorithm like Behavior Cloning. The comparison of ThriftyDAgger to other interactive imitation learning algorithms is easier to calibrate, since all algorithms are supplied with the exact same amount of offline supervisor actions and solicit supervisor actions online based on their respective policies. Note that 10,000 environment steps does not correspond to 10,000 samples, as IL algorithms generally only train on states that are labeled with expert actions.**
> > >
> > > I appreciate that the authors added a second simulated task, but the choice of stacking is a little unclear to me -- stacking seems like an easier task than peg insertion. Grasping the small box does not seem particularly difficult, and the precision needed to place the box on top of the large box seems pretty low as well. The low success rate of 5% for BC on 30 demonstrations also seems lower than expected - is there any intuition for why this might be the case?
> > >
> > > **While this task may not require higher precision manipulation than the peg insertion task, the initial poses of both blocks are randomized, while in the peg insertion task only the initial position of the ring is randomized (the target cylinder is in a fixed location). This results in a broad range of motions in the demonstrations, which is likely difficult for algorithms such as BC, which only use offline data, to generalize. Additionally, we note that this task, like the peg insertion task, is still relatively long horizon (70+ actions in most successful trajectories), making it challenging for imitation learning algorithms due to compounding approximation errors. Note that the peg insertion task was also chosen for its difficulty: it takes 100+ actions to complete successfully, and although the fitting may appear to be high tolerance, it is difficult for the robot to align in practice as the washer can easily get stuck against the edge of the peg.**

---

> > > ### Author Response · Authors · 2021-08-24
> > > **Response to Response (to Response), Part 2**
> > >
> > > Finally, the response mentioned that humans are not allowed to terminate the episode early - can you clarify why this is the case? It seems like a strange choice. Does this also mean that all episodes during data collection are the same in length?
> > >
> > > **Episodes are only terminated early if the goal is reached; in other words, successful episodes can be shorter in length, but all failures are the same length. The reason for this choice is to (1) make sure that the robot policy can observe how humans attempt to recover from challenging states and (2) remove human bias in deeming a state “unrecoverable,” which may be incorrect. This makes the human’s task consistent: take the best possible action in the given state. The tradeoff, as you suggest, is that the algorithm may collect human labels in irrecoverable regions that are not very useful. Future work can explore the effect of adding a “failure set” of states to the MDP (just as we have a goal set now).**

---

> > > > ### Comment · Reviewer_PRtc · 2021-08-28
> > > > **Thanks**
> > > >
> > > > Thanks for the clarifications - while I still think the Stacking task is actually easier than the Peg Insertion task, I appreciate the effort put into the additional experiments and responding to my questions, and will increase my score.

---

> > > > > ### Author Response · Authors · 2021-08-30
> > > > > **Reply**
> > > > >
> > > > > Thank you Reviewer PRtc, we really appreciate it.

---

### Official Review · Reviewer_pnHZ · 2021-07-23

**Originality:** Very Good
**Technical Quality:** Excellent
**Clarity Of Presentation:** Excellent
**Impact:** 3

**Recommendation:**

Strong Accept: I recommend accepting the paper and will argue for my recommendation even if other reviewers hold a different opinion.

**Summary:**

This paper presents ThriftyDAgger, an approach to reduce the burden on human operators during interactive imitation learning. To achieve this, the authors estimate both the novelty and the riskiness of states during execution---the latter being the most significant of their contributions---and only request human intervention when either prediction deems it necessary. The authors demonstrate good performance on a variety of simulated and real world task, achieving state-of-the art performance in both while simultaneously reducing the burden on human operators.

**Issues:**

I enumerate my suggested changes and provide justification for them in the "Strengths and Weaknesses" section above. In summary, they are as follows:

1. Inclusion of a clearer discussion of what is implied by the lower number of interventions for their technique and how those results should be interpreted to compare the different approaches.
2. Clarifying the discussion of the "Automatic Parameter Tuning" (including potentially changing the name of the relevant section and idea).
3. Some clearer language surrounding the theoretical motivation.

**Reviewer Expertise:**

Good: General knowledge of the area

**Strengths And Weaknesses:**


Overall, this is a solid, well-written paper and (despite a somewhat narrow theoretical contribution) the experiments are impressive and reinforce the utility and broad applicability of the technique. Among the experiments, perhaps the most impressive are those included in Table 2: Three-Robot Fleet Control User Study Results. These results do an excellent job of showing how human operators can simultaneously train a small fleet of agents while also trying to perform a task of their own. These results show the improvements afforded by ThriftyDAgger and were immediately convincing.

I have a number of comments and suggestions that, if addressed, would help improve the clarity of the work.

First, while it is certainly advantageous that ThriftyDAgger achieves a lower average number of interventions (for this section, I focus on the results shown in Table 1), it seems that the LazyDAgger has a similar "gating mechanism" to reduce the amount of context switching---and therefore the number of interventions. What do the authors hypothesize is the mechanism that allows for a lower number of interventions for ThriftyDAgger? (Is it perhaps what the authors refer to as the "automatic parameter tuning"?) Relatedly, how should we think about direct comparison between the different approaches for this "number of interventions" metric, since I expect it depends quite heavily on the parameter selection? My intuition is that simultaneously achieving lower numbers of interventions and also higher success rates is the key takeaway, but some clarity on the subject from the authors would be helpful.

Second, I understand why the authors use the phrase "Automatic Parameter Tuning" to describe how the risk and novelty are evaluated, yet I am not sure that this is a particularly appropriate name for this process since the parameters for this process are (technically) not changing, and instead the process by which novelty and risk are defined depend on statistics of the data. Instead, it might be more transparent to present the approach in this way, as the risk and novelty online as data is collected. The title of Sec 4.4 could be something like "Computing Risk and Novelty Thresholds from Data": more verbose, but also a better reflection of what the system is doing. In the text, the authors can still claim that this method avoids the need for oft-sensitive parameter tuning, even if one might not regard it as a "parameter tuning" approach. I am open to debate on the subject.

Finally, while I appreciate the definitions provided in Eqs (2) and (3), I am unsure that the definition in equation (3) comes up again, since this does not appear to be the optimization objective used to compute the expected policy. Could the authors clarify the relation between the process used to learn the policy and the objective defined in Eq. (3)? Similarly, while Eq. (2) is used to build intuition, the "burden" does appear to be directly calculated anywhere; this should be mentioned in Sec. 3.


**Summary Of Recommendation:**

This paper presents a novel idea that is particularly well-executed. Though the central contribution of the paper is not overwhelmingly transformative, they have put in considerable effort to clearly communicate their contribution in relation to other work and empirically demonstrate its benefits.

---

> ### Author Response · Authors · 2021-08-21
> **Response to Reviewer pnHZ**
>
> **Thank you for your time and feedback. We have addressed your comments below and uploaded a revision of the paper and supplement  with all the changes indicated in the responses below highlighted in blue.**
>
> “What do the authors hypothesize is the mechanism that allows for a lower number of interventions for ThriftyDAgger? (Is it perhaps what the authors refer to as the ‘automatic parameter tuning’?)”
>
> **Yes, by setting the switching thresholds online to achieve a target switching rate, we can directly control the expected number of interventions (Section 4.4). This allows us to achieve an intervention rate of approximately 1%, which is competitive with the rate achieved by prior algorithms without the need for extensive tuning.**
>
> “Relatedly, how should we think about direct comparison between the different approaches for this "number of interventions" metric, since I expect it depends quite heavily on the parameter selection? My intuition is that simultaneously achieving lower numbers of interventions and also higher success rates is the key takeaway, but some clarity on the subject from the authors would be helpful.”
>
> **Your intuition is correct, and we added more about this in Section 5.1. We want to show that ThriftyDAgger solicits human interventions more judiciously than prior algorithms, i.e. achieves a higher ratio of policy performance (both autonomous and intervention-aided) to the number of interventions. Simultaneously soliciting fewer interventions and achieving higher performance indicates that ThriftyDAgger is able to leverage the same amount of supervision better than the baselines (Table 1 and 2).**
>
> “The title of Sec 4.4 could be something like ‘Computing Risk and Novelty Thresholds from Data’: more verbose, but also a better reflection of what the system is doing.”
>
> **Thanks for the great suggestion. We have changed the title of the section as you suggested, and removed references to “automatic parameter tuning” throughout the text.**
>
> “Could the authors clarify the relation between the process used to learn the policy and the objective defined in Eq. (3)? Similarly, while Eq. (2) is used to build intuition, the ‘burden’ doesn’t appear to be directly calculated anywhere; this should be mentioned in Sec. 3.”
>
> **We use Equations (2) and (3) to build intuition and formalize the problem statement, but do not explicitly report supervisor burden as defined here. Instead, we report separated metrics that are more intuitive, such as the number of interventions and the total number of human actions used. In practice, Equation (3) is difficult to optimize exactly, so policies are learned by imitating aggregated data from the human supervisor, with supervisor burden controlled implicitly via the novelty and risk based switching criteria we present in the paper. We have updated Section 3 to make it clear that we do not explicitly optimize Equation (3), but instead present switching criteria which help limit supervisor burden to approximately satisfy the burden constraint in Equation (3), especially in environments with high latency (i.e. relative cost of a context switch to an individual action).**

---

### Official Review · Reviewer_UPFi · 2021-07-24

**Originality:** Good
**Technical Quality:** Very Good
**Clarity Of Presentation:** Very Good
**Impact:** 3

**Recommendation:**

Strong Accept: I recommend accepting the paper and will argue for my recommendation even if other reviewers hold a different opinion.

**Summary:**

This paper introduces another interactive learning method based on data aggregation, within the family of the DAgger algorithms. The method is able to control when there should be a switch between the autonomous control and the human teleoperation based on two different measures: uncertainty about the visited state, and probability of success which is computed similarly to a value function, assuming that there is a sparse reward only in a target state. The approach reduces the load of the teachers, since they don't have to keep full attention to the robot while it is learning, because the active learning scheme requests the new demonstrations according to the intervention criteria.

**Issues:**

See Strengths And Weaknesses

**Reviewer Expertise:**

Excellent: Expert knowledge on the topic of the paper

**Strengths And Weaknesses:**

- The paper is very easy to read, it is a pity not seeing the pseudocode in the main document, however, the text gives a clear explanation of its flow. The experiments cover well most of the questions that could come up about such a method. Below there are some comments with questions that can make the paper more complete.
- All this family of DAgger methods that have been developed incrementally (EnsembleDAgger, SafeDAgger, HG-DAgger, LazyDAgger), work under the strong assumption that the user is perfect and never makes mistakes. What happens when the teacher makes wrong demonstrations often? how is the performance of the learning process affected?
- Could it really be claimed that the novelty property is a contribution?
- How does the method behave at the beginning, when most of the states do not have a high Q value due to the failures of the learning policy, but the budget of human requests is limited, does it go to the extreme cases?
- How does the method deals with situations in which the human wants to intervene in order to change the current policy, but the robot is not switching to the teleoperation mode? It seems that a successful policy would not request feedback from the human, however, the method still should allow customizing the policies to the users' preferences.
What happens in tasks wherein the objective is not easily defined by the convergence to a set of target states? For instance, a biped robot learning to keep balance and walk?
- What kind of interface was used for the real robot experiment? what kind of considerations should be taken regarding the Human-Robot interface? These methods cannot be completely isolated from the human factors, therefore, these details are as important as the learning method implementation.
- Authors should elaborate the idea in line 236. It is not completely clear why negative results are discarded.
- The word "Berkeley" is really not needed to be mentioned, when describing the participants of the experiment. The review process is double-blind, however, this unnecessary information makes it clear what group the paper comes from. I really disliked this point.



**Summary Of Recommendation:**

The presentation of the method is clear, the experiments cover almost a complete evaluation of the proposed approach. There are many questions that should be approached, but I think the authors do not need to reconsider the method or the paper structure in order to address them.

---

> ### Author Response · Authors · 2021-08-21
> **Response to Reviewer UPFi**
>
> **Thank you for the valuable feedback. We have addressed your comments below and uploaded a revision of the paper and supplement with all the changes indicated in the responses below highlighted in blue. Due to space constraints our reply is split into two comments.**
>
> “What happens when the teacher makes wrong demonstrations often? How is the performance of the learning process affected?”
>
> **This is an interesting open question for the entire DAgger family of algorithms. Since we operate under similar assumptions to behavior cloning, DAgger, and the interactive baselines, we assume an optimal or near-optimal demonstrator. However an interesting direction for future work is outperforming a suboptimal demonstrator with reward learning, as in “Better-than-Demonstrator Imitation Learning via Automatically-Ranked Demonstrations” (Brown et al.) and “Learning from Suboptimal Demonstration via Self-Supervised Reward Regression” (Chen et al.).**
>
> “Could it really be claimed that the novelty property is a contribution?”
>
> **The novelty metric, by itself, is not novel; others have explored that metric previously. We revised the Contributions to more clearly state that the primary technical contributions are a new risk metric as well as switching criteria that synthesize the two metrics (see the Introduction).**
>
> “How does the method behave at the beginning, when most of the states do not have a high Q value due to the failures of the learning policy, but the budget of human requests is limited, does it go to the extreme cases?”
>
> **In the beginning of training, longer interventions are solicited, as the agent needs more help before it is confident enough to complete the task. This is reflected in Table 1 and Table 3, as the number of human actions falls significantly when using the same intervention criteria after training is complete (i.e. at execution time).**
>
> “How does the method deal with situations in which the human wants to intervene in order to change the current policy, but the robot is not switching to the teleoperation mode?”
>
> **In this work, we are interested in the case where there is limited human oversight available (for instance, in a robot call center with hundreds of robots but few supervisors), and there is a large benefit to having robots that can proactively request assistance. Although we compare against HG-DAgger, these methods are not necessarily mutually exclusive: there is no technical difficulty in adding human oversight to our method to allow this type of intervention. Whenever the robot is operating in autonomous mode, the human can simply choose to take control of the system. Studying the combination of human-gated and robot-gated interventions is another interesting direction for future work.**
>
> “What happens in tasks wherein the objective is not easily defined by the convergence to a set of target states? For instance, a biped robot learning to keep balance and walk?”
>
> **We acknowledge that our method would not apply to these tasks; in our problem statement, we note that the task must be specified as attempting to reach some goal set. We believe enough tasks can be formulated as goal-conditioned that our method is still broadly useful for robotics: for example, navigation to a goal, sequential bin-picking until a bin is empty, or folding cloth to a target configuration. Additionally, future work could explore reward learning (e.g. via preferences) to alleviate this assumption.**
>
> “What kind of interface was used for the real robot experiment? What kind of considerations should be taken regarding the human-robot interface?”
>
> **We clarified this and added a new figure in the revised version of the supplement (Figure 3, Section 7.3.3). Interventions for the real robot experiment are collected through a 7DOF teleoperation interface: the da Vinci Research Kit Master Tool Manipulator (MTM), which syncs state with the robot (the dVRK Patient Side Manipulator or PSM). We disable rotation of the end effector so that we effectively have 4DOF: 3D translation and opening or closing the gripper. This expert action is projected into the 2D plane before it is provided to the policy. The human-robot interface should be able to map the human action to the robot’s action space so that the human can directly provide examples for policy learning. Some care should also be taken in operating at a reasonable control frequency: for instance, if the end effector moves too much in the 1 second delay between image observations, the robot may be unable to imitate the action.**

---

> > ### Comment · Reviewer_UPFi · 2021-09-03
> >
> > Thanks to the authors for considering the reviews into the paper so it gets in an even better level. I would stick to my recommendation of acceptance

---

> ### Author Response · Authors · 2021-08-21
> **Response to Reviewer UPFi (Part 2)**
>
> “Authors should elaborate the idea in line 236.”
>
> **Thank you, we have done this in the revised paper (Section 5.1). The idea here is that since task failure is defined by reaching a timestep horizon, the metrics for failed episodes are skewed toward whatever horizon we set, which is arbitrary. For instance, a method with a lower success rate will appear to take more actions, as each failed episode will reach the task horizon. For completeness’ sake, we also added additional metrics for the total number of interventions, human actions, and robot actions (across successes and failures) as well as the number of successes and failures during training in Table 4-6 in the supplement.**

---

### Meta-Review · Area_Chair_JBUP · 2021-08-02

**Recommendation:** Accept (Oral)
**Confidence:** 4

**Metareview:**

This paper proposes an approach for generating queries for supervision from humans in an imitation learning context. The primary idea is to consider novelty of a state and the associated “risk” (i.e., likelihood of hindered task progress) in order to minimize context switching for the human supervisor. The resulting approach (ThriftyDagger) is demonstrated on simulation and hardware experiments involving manipulation tasks. In addition, results of a human user study on using the approach for controlling a group of three simulated robots is presented.

Strengths:
+ The reviewers agree that the inclusion of the risk term in deciding when to make queries to the human supervisor is novel.
+ The reviewers generally agree that the experimental evaluation of the approach is very thorough and convincingly demonstrates improvements in comparison to various baselines approaches.
+ The paper is very clearly written and the technical exposition is crisp.
+ The approach only requires choosing one hyperparameter (the context switching rate); other parameters are chosen based on this.

Weaknesses:
- The primary technical novelty of the approach is in the inclusion of the risk term. The reviewers request more justification and intuition for why this term helps the empirical performance (see the comments from Reviewer pnHZ and PRtc for more specific points to discuss).
- Reviewer PRtc noted that demonstrating the approach on a more challenging simulation task could be beneficial (especially since the performance gap between the proposed approach and baselines is smaller on the hardware experiments).

Suggestions:
I urge the authors to consider the reviewers’ detailed feedback in order to further improve the paper. The primary suggestion for improvement is to add more discussion and intuition on the benefits of including the risk term (which is the primary technical novelty in the approach). The reviewers also requested a few additional details and explanations, and suggested changing the phrase “automatic parameter tuning” to something more representative (e.g., “computing risk and novelty thresholds from data”).

----- Post rebuttal -----

The authors have addressed the reviewers' primary concerns during the rebuttal phase. In particular, the authors have (i) added intuition on why the risk term helps,  (ii) added a baseline (picking and stacking), and (iii) provided further details and clarifications. The revised paper is stronger as a result, and the reviewers are in agreement that the paper constitutes a strong contribution and should be accepted.

---

> ### Author Response · Authors · 2021-08-21
> **Response to Area Chair JBUP**
>
> **We thank the Reviewers and Area Chair for their feedback and address each comment below. We have uploaded a revision of the paper and supplement with all the changes indicated in the responses below highlighted in blue.**
>
> **We are encouraged by the high ratings in originality, technical quality, and clarity of presentation. Regarding the impact score, we believe that ThriftyDAgger could have relatively near-term impact in practice, as commercial robots are being employed at scale with increasing regularity in real-world applications such as driving, warehousing, manufacturing. For example, autonomous driving company Zoox now has a TeleGuidance system, Nimble Robotics has remote operators to handle edge cases in robot bin-picking, and Plus One Robotics offers Yonder, software for supervising robot fleets. When deployed in the real world, even relatively reliable robotic policies encounter new situations and must fall back on human expertise. Thus, the ability to effectively and efficiently seek human interventions in these situations is critical for deploying learning-based robotic systems reliably and scalably in the real world. To clarify this, we have revised the Introduction.**
>
> The primary technical novelty of the approach is in the inclusion of the risk term. The reviewers request more justification and intuition for why this term helps the empirical performance (see the comments from Reviewer pnHZ and PRtc for more specific points to discuss).
>
> **Yes, thank you for pointing this out. In many tasks there exist states that are familiar to the robot that nevertheless have a low probability of success and should result in a request for assistance: regions with very low tolerance for error, for instance. Moreover, the robot should only take back control from the human when states are both familiar (low novelty) and have a high probability of success under the robot policy (low risk). We have updated Sections 4.2 and 4.3 with more intuition and describe the intuition in more detail in our responses to Reviewers pnHZ and PRtc.**
>
> Reviewer PRtc noted that demonstrating the approach on a more challenging simulation task could be beneficial (especially since the performance gap between the proposed approach and baselines is smaller on the hardware experiments).
>
> **We have added an additional simulation domain as requested (Section 7.3.2 and Table 5 in the supplement) and discuss further details on these results in the response to Reviewer PRtc. In this task, a robot must pick up and stack a cube from a randomized initial pose on top of another cube in another randomized pose. The geometric symmetry of the cubes, randomized start and goal, and small area for block placement make this a difficult task where Behavior Cloning achieves only 5% success. Here, as in the other domains, ThriftyDAgger matches the performance of HG-DAgger and outperforms the other baselines and ablations.**
>
> **A related point is the relationship between ThriftyDAgger and HG-DAgger. In physical experiments, although ThriftyDAgger performs very similarly to HG-DAgger, we emphasize that even achieving competitive performance with HG-DAgger is a surprising result, as HG-DAgger is like an oracle in that it requires a human operator to decide the timing and length of all interventions. Comparable performance to HG-DAgger indicates that ThriftyDAgger is able to learn intervention criteria of similar quality to a human expert with detailed knowledge about the task. Accordingly, we have reduced emphasis on outperforming HG-DAgger in the paper.**
>
> The reviewers also requested a few additional details and explanations, and suggested changing the phrase “automatic parameter tuning” to something more representative (e.g., “computing risk and novelty thresholds from data”).
>
> **We have included all requested explanations in the responses below and added necessary details both to the responses and the paper/supplementary material. We agree that Computing Risk and Novelty Thresholds from Data is a better name and have updated this accordingly.**

---

### Decision · Program_Chairs · 2021-09-13

**Decision:**

Accept (Oral)

**Comment:**

This paper proposes an approach for generating queries for supervision from humans in an imitation learning context. The primary idea is to consider novelty of a state and the associated “risk” (i.e., likelihood of hindered task progress) in order to minimize context switching for the human supervisor. The resulting approach (ThriftyDagger) is demonstrated on simulation and hardware experiments involving manipulation tasks. In addition, results of a human user study on using the approach for controlling a group of three simulated robots is presented.

Strengths:
+ The reviewers agree that the inclusion of the risk term in deciding when to make queries to the human supervisor is novel.
+ The reviewers generally agree that the experimental evaluation of the approach is very thorough and convincingly demonstrates improvements in comparison to various baselines approaches.
+ The paper is very clearly written and the technical exposition is crisp.
+ The approach only requires choosing one hyperparameter (the context switching rate); other parameters are chosen based on this.

Weaknesses:
- The primary technical novelty of the approach is in the inclusion of the risk term. The reviewers request more justification and intuition for why this term helps the empirical performance (see the comments from Reviewer pnHZ and PRtc for more specific points to discuss).
- Reviewer PRtc noted that demonstrating the approach on a more challenging simulation task could be beneficial (especially since the performance gap between the proposed approach and baselines is smaller on the hardware experiments).

Suggestions:
I urge the authors to consider the reviewers’ detailed feedback in order to further improve the paper. The primary suggestion for improvement is to add more discussion and intuition on the benefits of including the risk term (which is the primary technical novelty in the approach). The reviewers also requested a few additional details and explanations, and suggested changing the phrase “automatic parameter tuning” to something more representative (e.g., “computing risk and novelty thresholds from data”).

----- Post rebuttal -----

The authors have addressed the reviewers' primary concerns during the rebuttal phase. In particular, the authors have (i) added intuition on why the risk term helps,  (ii) added a baseline (picking and stacking), and (iii) provided further details and clarifications. The revised paper is stronger as a result, and the reviewers are in agreement that the paper constitutes a strong contribution and should be accepted.